# Detection of Mycotoxins in Cereal Grains and Nuts Using Machine Learning Integrated Hyperspectral Imaging: A Review

**DOI:** 10.3390/toxins17050219

**Published:** 2025-04-27

**Authors:** Md. Ahasan Kabir, Ivan Lee, Chandra B. Singh, Gayatri Mishra, Brajesh Kumar Panda, Sang-Heon Lee

**Affiliations:** 1UniSA STEM, University of South Australia, Mawson Lakes 5095, Australiasang-heon.lee@unisa.edu.au (S.-H.L.); 2Department of Electronics and Telecommunication Engineering, Chittagong University of Engineering and Technology, Chittagong 4349, Bangladesh; 3Advanced Post Harvest Technology Centre, Lethbridge College, Lethbridge, AB T1K 1L6, Canada; 4Agricultural and Food Engineering Department, Indian Institute of Technology Kharagpur, Kharagpur 721302, West Bengal, India

**Keywords:** mycotoxins, aflatoxin, deoxynivalenol, fumonisins, hyperspectral imaging, machine learning, deep learning, mycotoxins detection

## Abstract

Cereal grains and nuts are the world’s most produced food and the economic backbone of many countries. Food safety in these commodities is crucial, as they are highly susceptible to mold growth and mycotoxin contamination in warm, humid environments. This review explores hyperspectral imaging (HSI) integrated with machine learning (ML) algorithms as a promising approach for detecting and quantifying mycotoxins in cereal grains and nuts. This study aims to (1) critically evaluate current non-destructive techniques for processing these foods and the applications of ML in identifying mycotoxins through HSI, and (2) highlight challenges and potential future research directions to enhance the reliability and efficiency of these detection systems. The ML algorithms showed effectiveness in classifying and quantifying mycotoxins in grains and nuts, with HSI systems increasingly adopted in industrial settings. Mycotoxins exhibit heightened sensitivity to specific spectral bands within HSI, facilitating accurate detection. Additionally, selecting only relevant spectral features reduces ML model complexity and enhances reliability in the detection process. This review contributes to a deeper understanding of the integration of HSI and ML for food safety applications in cereal grains and nuts. By identifying current challenges and future research directions, it provides valuable insights for advancing non-destructive mycotoxin detection methods in the food industry using HSI.

## 1. Introduction

Cereal grains and nuts are the most produced and consumed foods all over the world due to their high nutritional value. In warm and humid climates, a variety of fungi can flourish in cereal grains and nuts from cultivation to storage. The fungi can produce numerous mycotoxins through their secondary metabolism [1]. Mycotoxins are toxic, mutagenic, and carcinogenic compounds that pose substantial health risks to humans and animals [2]. *Aspergillus* and *Fusarium* are two fungi that are frequently detected in nuts and cereal grains. *Aspergillus* species produce various life-threatening biotoxins such as aflatoxins (AFs), ochratoxins (OTAs), patulin (PAT), and citrinin (CIT). Aflatoxin B1 is the most dangerous mycotoxin among the four main aflatoxins (B1, B2, G1, and G2) produced by *Aspergillus* [3]. The major mycotoxins that *Fusarium* fungi produce are trichothecenes, fumonisins (FM), and zearalenone (ZEN) [4]. These mycotoxins have been associated with hormone-related disorders and may contribute to cancer development in humans [5]. According to the International Agency for Research on Cancer (IARC), fumonisins are classified as possibly carcinogenic to humans (Group 2B), while others such as zearalenone (ZEN), deoxynivalenol (DON), and T-2 toxin are classified as not classifiable as to their carcinogenicity to humans (Group 3) [6]. Besides that, several hundred different mycotoxins have been identified in cereal grains and nuts, which impact human health and livestock. The most commonly found mycotoxins that contaminate cereal grains and nuts include aflatoxins, ochratoxin A, patulin, T-2 toxin, fumonisins, zearalenone, ergot alkaloids, *Alternaria* toxins, and deoxynivalenol (DON) [7]. The most common mycotoxins that are found in cereal grains and nuts and their possible health threats are tabulated in Table 1.

Mycotoxin contamination is the greatest threat to food and one of the common causes of food poisoning and spoilage. According to the World Health Organization (WHO), around 600 million people become ill from eating contaminated food every year, and around 4.2 million die [23]. Also, contaminated food accounts for USD 110 billion in economic and health losses each year and builds trade barriers in the global market. According to the Food and Agricultural Organization (FAO), about 25% of the world’s crops are contaminated by mycotoxin-producing fungi. Therefore, a fast and reliable framework is required to handle this threat, which can detect and quantify the mycotoxin at a maximum acceptable level from the complex food materials [24]. Mycotoxin detection and quantification procedures must be extremely sensitive, and the capability should be as low as a few parts per billion (ppb), as different nations have varying mycotoxin regulation limitations. However, most current mycotoxin detection and quantification methods are destructive, time-consuming, and costly, and they face challenges related to sampling variability caused by the uneven distribution of contamination within bulk samples. Thus, they do have some limitations [25]. Different tests for the same batch produce different findings because of the sampling issue [26]. As a result, there is a greater need for more efficient and reliable procedures in industrial food processing applications.

The commonly used mycotoxin detection methods are gas chromatography, high-performance liquid chromatography (HPLC), thin-layer chromatography (TLC) [27], rapid test kits, enzyme-linked immunosorbent assay (ELISA) [28], and molecular identification techniques (fluorescence property) [29]. Among these, ELISA and HPLC provide high accuracy and selectivity, and their detection level is as low as 0.01 ppb. Though rapid test kits like ELISA, immunoassay, and membrane flow techniques have a satisfactory quantification capability [30], these rapid kits are only for single use. On the other hand, the ELISA method has sensitivity constraints on the antibodies, which limit the detection range of this method. Besides that, sample preparation is another major problem of the mentioned methods. Some samples may be contaminated at a higher level, and there is a high possibility of random distribution of contaminated samples in the lot. Therefore, the highly contaminated samples may not be selected equally for analysis at each time [31]. Also, the mentioned methods are fast but require some time, which makes them impractical for rapid in-line industrial applications. In-line food safety assessment in the industry requires fast, non-destructive, and cost-effective methods [32].

Over the past few decades, the food processing industry has employed computer vision systems with spectroscopic techniques to detect and quantify diverse contamination in cereal, nuts, and their products [33]. The RGB imaging, X-ray, fluorescence spectroscopy, visible near-infrared (Vis-NIR), and NIR hyperspectral imaging are the primarily used imaging approaches to identify different mycotoxins in cereal grains and nuts, combined with a machine learning algorithm [34]. RGB images make it simple to recognize the external characteristics of foods, such as the size, shape, and texture. However, the RGB images cannot provide any chemical information [35].

Hyperspectral imaging (HSI) is the most promising and powerful imaging approach for the detection of mycotoxins using machine learning and can describe the chemical properties of agricultural foods and food products. This technique, coupled with machine learning (ML) algorithms, is more widely used to identify the mycotoxins in cereal grains and nuts such as almonds [36], peanuts [37], pistachios [38], wheat [39], and corn [40]. However, the amount of data in a hyperspectral image is enormous, and a traditional statistical classifier cannot handle them. In contrast, ML algorithms can learn from complex data. The advancement of ML algorithms opens a way to handle HSIs more conveniently. However, the efficiency and effectiveness of an ML algorithm depend on the learning algorithm’s performance and the characteristics of the data. The spectral responses of various cereal grains and nuts are different due to their complex nutritional structure. Therefore, selecting a suitable ML algorithm for cereal grains and nuts is challenging. The reason is that different learning algorithms have different purposes, and similar types of learning algorithms may provide different outcomes due to the characteristics of the data. Therefore, several literature reviews have been conducted on the spectroscopic techniques coupled with the ML algorithm on food safety applications [36,41]. There are limited reviews of ML algorithms to detect and quantify mycotoxin in cereal grains and nuts using hyperspectral images. Sendin et al. reviewed NIR HSI techniques to detect fungi-contaminated cereals [41]. Wu et al. reviewed the use of different spectroscopic techniques for detecting fungi and aflatoxin in nuts and dried fruits [42]. Optical imaging and spectroscopic methods for mold and mycotoxin detection in grain and nuts were reviewed by [25]. Nevertheless, only a few evaluations of ML methods that use hyperspectral images to identify and measure mycotoxin in nuts and cereal grains have been conducted.

This work comprehensively analyzes ML methods for utilizing HSI to detect and quantify mycotoxins in cereal grains and nuts. It identifies and evaluates the most effective ML techniques for processing and interpreting high-dimensional spectral data, offering practical guidance for researchers and practitioners developing non-destructive food safety solutions. This review synthesizes findings from a wide range of studies, presents comparative insights into algorithm performance across different applications, and discusses real-world challenges such as data variability and contamination heterogeneity. In addition, this paper outlines current advancements, highlights emerging research opportunities, and defines key technical and practical challenges that remain in the field.

## 2. Search Methodology

This literature review was conducted to assess the application of HSI combined with ML for mycotoxin detection in cereal grains and nuts. As the number of publications in this field has grown rapidly in recent years, the review process involved a comprehensive approach to capture the most relevant studies while maintaining a structured methodology. The methodology followed four distinct steps: (1) review planning, (2) conducting the review, (3) data screening, and (4) result analysis.

### 2.1. Review Planning

In the planning phase, the objectives, scope, and criteria for article selection were clearly defined to ensure a rigorous and reproducible review process. The primary objective was to examine how machine learning algorithms have been applied to detect mycotoxins in cereal grains and nuts using HSI. The scope focused on research conducted between 2015 and 2024, primarily in English-language articles. The inclusion criteria were articles that developed or used machine learning models to detect mycotoxins in these matrices using hyperspectral images. Exclusion criteria included studies focused on remote sensing, grain-tree- or nut-tree-related papers, and literature reviews.

### 2.2. Conducting the Review

This review was conducted by systematically gathering literature from the following academic databases: Scopus, ScienceDirect, Springer, and Google Scholar. A variety of keyword combinations, including ‘mycotoxin’, ‘hyperspectral imaging’, ‘cereals grins’, ‘nuts’, and ‘machine learning’, were used to ensure comprehensive coverage of the topic.

### 2.3. Data Screening

The articles obtained from the initial search were screened based on predefined inclusion/exclusion criteria. The search yielded 83 relevant papers. The selection process involved filtering articles by imaging techniques, study design, and language. Non-experimental papers (e.g., reviews) and studies that were not directly related to mycotoxin detection in cereal grains and nuts were excluded. The focus on cereal grains and nuts was applied as an inclusion/exclusion criterion during the screening stage, meaning that while the initial search string did not explicitly restrict results to these matrices, the focus was applied during screening based on relevance to the review objectives.

### 2.4. Result Analysis

Following data screening, the selected articles were analyzed to assess the effectiveness of the machine learning algorithms used for mycotoxin detection in cereal grains and nuts. This phase involved interpreting the findings from the papers, analyzing trends in the data, and evaluating the performance of the various algorithms. The most recent and relevant studies were chosen for further discussion in the subsequent sections.

## 3. Hyperspectral Analysis of Mycotoxins in Cereal and Nuts

Cereal grains and nuts are commonly infected by mold, which creates various mycotoxins in the later stages. Some mycotoxins are highly toxic and carcinogenic, and certain types cannot be eliminated from food through heating, cooking, or other conventional processing methods. Continuous consumption of these mycotoxins poses a significant health risk to both humans and animals. The following subsections discuss the hyperspectral responses and various approaches for detecting mycotoxins in cereal grains and nuts.

### 3.1. Hyperspectral Response of Different Mycotoxins

HSI provides distinct spectral characteristics in multiple wavelengths to detect mycotoxins in cereals and nuts. Key wavelengths were identified in the visible–near infrared (Vis-NIR) and shortwave infrared (SWIR) regions, which correlate with the chemical properties of different mycotoxins.

HSI has proven to be an effective method for detecting aflatoxin B1 (AFB1) in various agricultural products by identifying key spectral bands. Studies have demonstrated that specific wavelengths are highly predictive of AFB1 contamination. For instance, wavelengths at 956, 984, 1046, 1149, 1317, 1400, 1459, and 1729 nm were found to be particularly significant for AFB1 detection [32,36,43]. Additionally, [44] emphasized the role of the 1200 nm wavelength, which is associated with the N-H bonds in amino acids and the benzene rings in AFB1, further confirming its utility for AFB1 detection in cereals.

For detecting DON in wheat, hyperspectral imaging has a significant response at wavelengths of 1408 nm, 1904 nm, and 1919 nm, especially at higher concentrations [45]. These bands capture changes in starch and protein content caused by *Fusarium* infections, which are related to DON contamination. The wavelengths 1300, 1350, and 1480 nm have proven essential for detecting OTA in wheat [46]. Wavelengths 1300 and 1350 nm correspond to fungal infection and carbohydrate presence, the 1480 nm wavelength is particularly significant for identifying protein content and OTA contamination. The most characteristic wavelengths for T-2 toxin, identified from the regression coefficients, were 1038, 1110, and 1393 nm [47].

The FM is detectable using hyperspectral imaging techniques across the Vis-NIR and SWIR regions. The SWIR showed higher peaks and valleys at 1320, 1870, and 2254 nm and 1208, 1474, and 1940 nm, respectively [48]. The absorption at wavelengths 1208 nm (C-H stretching) [49], 1474, 1870 nm (O-H vibrations associated with starch) [50], and 1895 nm (C-O-H and C-O-C deformations) [51] are significant for identifying fumonisin contamination in cereals.

### 3.2. Detection of Mycotoxins in Wheat

Wheat is one of the largest-grown and -consumed cereal grains worldwide. Also, it is susceptible to a toxin-producing fungal strain. These toxic compounds are produced by molds such as *Fusarium*, *Aspergillus*, and *Alternaria* species. *Fusarium* head blight (FHB) is a fungal-infected disease, and wheat is commonly affected by FHB. The FHB produces mycotoxin by secondary metabolism, especially DON [52]. The DON mycotoxin mainly infects the wheat’s organic properties like carbohydrates, proteins, and lipids [53]. Besides that, wheat is commonly contaminated with ZEN, AFB1, and OTA mycotoxins.

In recent years, several studies have employed HSI integrated with ML techniques to detect and quantify mycotoxins such as DON, OTA, and aflatoxin B1 in wheat, as summarized in Table 2. These approaches leverage specific spectral bands and advanced preprocessing techniques to enhance detection accuracy. Various spectral ranges have been explored specifically for detecting DON in wheat. Femenias et al. investigated the 900–1700 nm range using MSC, SG smoothing, first and second derivatives, and SNV preprocessing and applied LDA, NB, and KNN [54,55]. These models achieved high classification accuracies, with LDA and KNN models obtaining 100% accuracy. Similarly, [39] applied KNN classifiers in the 960–1700 nm range, achieving 80% accuracy in detecting DON in Canadian Western Red Spring wheat, demonstrating the potential of HSI and KNN for rapid, reliable DON detection.

Other studies, such as Liang et al., compared detection methods within the 400–2500 nm range, using SNV and MSC preprocessing with ML models like SVM and SAE [58]. Their models achieved an accuracy of 96% for DON classification in wheat samples. Similarly, Shen et al. applied PLSR and SVR models using the 970–1623 nm range and achieved a coefficient of regression (R^2^) value of 0.88, further confirming the effectiveness of HSI combined with advanced ML models for DON detection [49].

For aflatoxin B1 detection, Jiang et al. developed a PLSR model using wavelengths between 899 and 1725 nm, with second derivative and SNV preprocessed data. The model obtained a high R^2^ value of 0.9935, illustrating the model’s robust performance in quantifying aflatoxin B1 in wheat [60]. Senthilkumar et al. employed the 1000–1600 nm range and developed a discriminant analysis (DA) model for OTA detection [61]. Their models achieved remarkable classification accuracies (Acc) of 99.8% and 100%, respectively, showcasing the reliability of HSI with ML techniques for detecting OTA contamination in wheat.

### 3.3. Detection of Mycotoxins in Corn

Corn is one of the highly consumed cereal grains and grows throughout the world. For mycotoxin detection in corn, various studies have utilized HSI within specific wavelength ranges, applying a combination of preprocessing methods and ML models to enhance accuracy. The summarized key findings to detect mycotoxin in corn using HSI are tabulated in Table 3.

AFB1 detection has been widely explored across multiple spectral ranges, with 327–2500 nm being the most effective. Tian et al. used PLSR within the 327–1098 nm and 930–2548 nm ranges, achieving a high R^2^ value of 0.924 [64]. Similarly, Zhou et al. employed LDA on the 1100–2000 nm range, achieving an accuracy of 95.56% by focusing on critical spectral features [69]. In contrast, Gao et al. achieved an optimal accuracy of 99.38% with RF and KNN models using the 292–865 nm range with MSC and PCA preprocessing [70]. Other approaches, such as PLS-DA and SVM on 900–2500 nm with SNV and SG preprocessing, consistently provided a high accuracy of 95.7% [48]. These findings indicate that both narrow and broad spectral ranges can be highly effective with robust preprocessing and feature selection.

DON detection has shown effectiveness in the 893–1730 nm spectral range with MSC, SNV, and SG preprocessing techniques. Borràs-Vallverdú et al. reported an accuracy of 98.6% with RF and neural network classifiers [71]. Additionally, Parrag et al. achieved an R^2^ of 0.974 and a 98.8% accuracy using PLSR and PLS-DA within the 900–1700 nm range [72]. These methods illustrate the potential of combining comprehensive preprocessing techniques with ensemble models to improve DON detection accuracy, particularly for complex corn samples.

To detect FM, the 1000–2100 nm range proved effective with SNV preprocessing and PLSR, yielding an R^2^ of 0.98 [73]. This suggests a strong correlation of FM spectral features within the NIR region. ZEN detection was performed within 400–1000 nm, achieving an accuracy of 93.33% using various models, including SVM, DA, PLSR, and neural networks [74]. The success of these models highlights the ability of HSI, combined with appropriate ML techniques, to detect a range of mycotoxins in corn.

### 3.4. Detection of Mycotoxin in Other Cereals and Grains

Food processing is essential for cereal grains and crops to reduce mycotoxin levels and ensure compliance with regulatory safety limits. The parametric literature review of mycotoxin detection in various cereal grains is tabulated in Table 4.

To detect DON contamination in barley, Fan et al. applied a CNN on HSI data in the 367–1048 nm range, achieving an accuracy of 89.81% [75]. Similarly, Su et al. employed a partial least squares discriminant analysis (PLSDA) with wavelengths ranging from 382 to 1030 nm for detecting DON, achieving an R^2^ of 0.931 [76]. For OTA detection in barley, Senthilkumar et al. developed a DA model in the 1000–1600 nm range, yielding a perfect classification accuracy of 100% [77].

Focusing on DON and T-2 toxin in oats, Teixido-Orries et al. developed models in the 900–1700 nm spectral range, using preprocessing techniques such as SNV and first derivatives [47]. The author tested multiple ML models, including PLSR, RF, KNN, and NB. Their model for T-2 toxin detection achieved a high accuracy of 94.1%, while DON detection in oats reached an accuracy of 77.8%. In a separate study, Tekle et al. used normalization and PLSR with the 1000–2500 nm range, achieving an R^2^ value of 0.8 for DON in oats [78]. This review suggests that CNN and PLDA perform particularly well for DON detection in barley, with CNN showing superior accuracy. For DON and T-2 toxins in oats, the combination of SNV, first derivatives, and PLSR/RF appears effective.

### 3.5. Detection of Mycotoxins in Peanuts

Peanut is a nut that contains 30% protein and 50% oil; it also contains vitamin E, resveratrol, and other important health-improving substances [79]. The consumption of peanuts reduces the possibility of cardiovascular and inflammatory diseases [80]. Much like peanuts, almonds, pistachios, and other nuts are sought after by consumers due to their excellent nutritional value. However, nuts are also affected by fungi in a natural way, which can produce mycotoxins through a secondary metabolism process. The peanut, almond, and pistachio industries rely on HSI and ML techniques to detect mycotoxin contamination, particularly AFB1, due to its high toxicity and carcinogenic nature. The following overview summarizes recent approaches, wavelengths, and ML methods used to quantify mycotoxin in these nuts, as detailed in Table 5.

The use of CNN and PLSR has shown high accuracy across different spectral ranges to detect AFB1 in peanut kernels. Zhu et al. used CNNs (e.g., ResNet18 and AlexNet) in the 292–865 nm and 415–799 nm ranges, achieving R^2^ values of 0.88 and a classification accuracy of 91.3% [81,82]. This approach highlights CNNs’ strength in handling complex data, especially for pixel-level analysis. Alternatively, Guo et al. used traditional regression models, including PLS and principal component regression (PCR), over a broader 900–2500 nm range [83]. This achieved 100% classification accuracy, indicating that broader spectral ranges, with simpler ML models like PLS, can provide robust classification with effective feature extraction. For high specificity, He et al. and Tao et al. demonstrated that SVM and PLSDA, respectively, achieved high accuracy (up to 94.8%) with targeted wavelengths around 400–1000 nm [84]. This suggests that SVM and PLSDA could be more efficient when computational simplicity and specificity are prioritized.

For almonds, which also face a significant aflatoxin contamination risk, studies using 900–1700 nm with SVR, Gaussian process regression (GPR), and PLSR yield excellent results. Kabir et al. achieved R^2^ values of 0.966 using SVR and GPR, indicating that SVR excels with broad spectral data [85]. Additionally, Mishra et al. applied PLSR and multiple linear regression (MLR) within the same range, achieving similar precision (R^2^ of 0.958), showing that for quantification tasks, regression models are particularly effective in identifying aflatoxin in almonds with high reliability [36].

Aflatoxin B1 contamination is also a concern in pistachios, with studies suggesting high efficacy with wider spectral ranges. Wu and Xu employed LDA and sparse multivariate linear regression (SMLR) within 694–988 nm, reaching a 90% accuracy [38], while Wu et al. used SVM and PLSR within 408–1007 nm, achieving an impressive 98% accuracy [86].

**Table 5 toxins-17-00219-t005:** ML techniques to detect mycotoxin in nuts using hyperspectral images.

Nuts	Mycotoxin	Sample Number	Wavelength (nm)	Preprocessing	ML Methods	Performance	Reference
Peanut	AFB1	20	415–799	Noise fraction	CNN	Acc: 91.3%	[82]
Peanut	AFB1	1260	900–2500	SG, 1st dev	PLS, PCR	Acc: 100%	[83]
Peanut	AFB1	73	292–865	N-FINDR	ResNet18, LeNet, AlexNet	R^2^: 0.88	[81]
Peanut	AFB1	150	1000–2500	SNV, 1st and 2nd dev	SVR, PLSR	R^2^: 0.95	[37]
Peanut	AFB1	150	400–1000	SG, SNV, 1st dev, MSC	PCA-LDA, SVM	Acc: 93%	[84]
Peanut	AFB1	250	400–720	RI, DRI, RRI, NDRI	SVM	Acc: 93.1%	[87]
Peanut	AFB1	210	400–2500	SNV	PLS-DA	Acc: 94.8%	[88]
Peanut	AFB1	73	292–865	Raw Image	CNN,	Acc: 95%	[89]
Peanut	AFB1	13,000	415–799	PCA	DCNN	Acc: 97.87%	[90]
Almond	AFB1	400	900–1700	SNV, SG, 1st and 2nd dev	SVM, LR, LDA, QDA	Acc: 95%	[91]
Almond	AFB1	500	900–1700	SNV, SG, 1st and 2nd dev	RF, QDA	ACC: 96. 37%	[92]
Almond	AFB1	1520	900–1700	SNV, SG, 1st and 2nd dev	ResNet, InceptionNet, Inception ResNet	ACC: 95.91%	[93]
Almond	AFB1	1832	900–1700	SNV, SG, 1st and 2nd dev	SVM, Subspace, Rusboost	ACC: 96%	[94]
Almond	AFB1	3596	900–1700	Raw Image	3D CNN	ACC: 90.81%	[95]
Almond	AFB1	500	900–1700	SNV, SG, 1st and 2nd dev	SVR, GPR	R^2^: 0.966	[85]
Almond	AFB1	936	900–1700	SNV, SG, 1st and 2nd dev	SVM	Acc: 98.7%	[96]
Almond	AFB1	465	900–1700	SNV, SG, 1st and 2nd dev	PLSR, MLR	R^2^: 0.958	[36]
Pistachio	AFB1	300	694–988	SNV, SG	LDA, SMLR	Acc: 90%	[38]
Pistachio	AFB1	300	408–1007	SG	SVM, PLSR	Acc: 98%	[86]
Pistachio	AFB	300	400–1000	Raw Image	ResNet	Acc: 96.67%	[97]

A**F**B1 (aflatoxin B1); SNV (standard normal variate); MSC (multiplicative scatter correction); SG (Savitzky–Golay); LDA (linear discriminant analysis); SVM (support vector machine); PLSR (partial least squares regression); SVR (support vector regression); GPR (Gaussian process regression); MLR (multiple linear regression); KNN (k-nearest neighbors); RF (random forest); CNN (convolutional neural network); QDA (quadratic discriminant analysis); SMLR (sparse multivariate linear); PCR (principal component regression).

## 4. ML Techniques for Detection of Mycotoxin in Cereals and Nuts Using HSI

Machine learning algorithms play a crucial role in analyzing and classifying cereals and nuts based on their physicochemical properties and spectral data. The following section provides a detailed analytical overview of various supervised machine learning algorithms used in recent studies, discussing their applications and performance in different cereal and nut classifications. The percentage usage of the most common ML algorithms for mycotoxin detection using HSI is illustrated in Figure 1. The most frequently used ML algorithms include PLSR, DA, KNN, and SVM. However, other supervised ML algorithms, such as RF, NB, artificial neural networks (ANNs), decision trees (DT), and MLR, are also widely utilized. A visualization of some of these ML algorithms is presented in Figure 2.

### 4.1. Partial Least Squares Regression

Partial least squares regression is widely utilized for mycotoxin detection in agricultural products due to its ability to process high-dimensional and collinear spectral data effectively. By projecting both spectral data and mycotoxin concentration levels into a lower-dimensional latent space, PLSR enhances the extraction of relevant spectral features while reducing data dimensionality. This makes it particularly effective for HSI-based mycotoxin quantification, as it mitigates noise and addresses overlapping spectral bands. PLSR has demonstrated a strong predictive performance in detecting aflatoxins and fumonisins in maize and corn, where its ability to capture critical spectral variations has led to the accurate classification of contaminated samples [72,98]. Similarly, its application in peanut classification has provided rapid and non-destructive screening for aflatoxin contamination, reinforcing its practicality for food safety monitoring [37,83]. Additionally, studies on wheat and barley highlight PLSR’s effectiveness in distinguishing between contaminated and non-contaminated grains based on spectral signatures linked to chemical composition [78,99]. However, selecting the optimal number of latent variables remains a critical challenge—too few can lead to underfitting, while too many may introduce noise and reduce model robustness. PLSR is a powerful tool for mycotoxin quantification when sufficient training data are available and the contaminant’s spectral features are well defined within the HSI range, allowing for precise and reliable detection.

### 4.2. Multiple Linear Regression

Multiple linear regression is a statistical method frequently used for mycotoxin detection in HSI due to its interpretability and ability to model linear relationships between spectral features and contamination levels. MLR has been applied in maize and corn classification to predict aflatoxin and fumonisins contamination with notable accuracy [100]. It has also been used in peanut classification to model chemical variations caused by mycotoxin-producing fungi [84]. While MLR assumes linearity and is sensitive to multicollinearity and outliers, these challenges can be mitigated through feature selection and data transformations to enhance model reliability.

### 4.3. Support Vector Machine

Support vector machine is another widely used algorithm for mycotoxin detection due to its ability to handle high-dimensional and nonlinear data. By transforming data into a higher-dimensional space, SVM constructs an optimal hyperplane that maximizes the separation between contaminated and non-contaminated samples, reducing classification errors. Kernel functions, such as the radial basis function (RBF) and polynomial kernels, help capture complex spectral patterns associated with mycotoxins. SVM has demonstrated high accuracy in detecting mycotoxin contamination in maize and corn using HSI [63,67,101] and has been effectively applied to wheat classification for identifying fungal infections linked to mycotoxin production [49,58]. Additionally, SVM has proven reliable for aflatoxin detection in peanuts, enabling the precise identification of contaminated samples [37,84]. However, SVM’s performance depends on selecting optimal kernel parameters and managing sensitivity to outliers. These challenges can be mitigated through cross-validation, regularization techniques, and effective spectral data preprocessing, such as normalization and noise reduction.

### 4.4. Discriminant Analysis

The discriminant analysis classifier leverages spectral features from hyperspectral and multispectral imaging to differentiate contaminated from non-contaminated samples. By projecting data into a lower-dimensional space, DA reduces complexity while identifying key variables that contribute to class separation. It employs functions such as linear discriminant analysis (LDA) and quadratic discriminant analysis (QDA) to classify new observations based on predefined groups. LDA, the most common variant, assumes a Gaussian distribution for class observations with a shared covariance matrix, allowing for efficient classification with linear decision boundaries. DA has been successfully applied in maize and corn classification to distinguish mycotoxin-contaminated samples [100,102], as well as in wheat classification to detect fungal contamination affecting mycotoxin levels [57,58]. Additionally, it has been used in peanut classification for identifying aflatoxin-contaminated samples [84,88]. The effectiveness of DA depends on the choice of discriminant functions and the quality of training data, ensuring accurate classification across different agricultural products.

### 4.5. Decision Tree

Decision tree algorithms are used for mycotoxin detection in cereals and nuts due to their interpretability and ability to model complex decision boundaries in hyperspectral imaging data. A DT continuously splits data from root to leaf nodes based on attribute values, commonly using Gini impurity or entropy for decision making, while DT models effectively classify mycotoxin-contaminated samples in maize, corn, and peanuts [81,103,104]. They are prone to overfitting and sensitive to minor data variations. These limitations can be mitigated through pruning techniques, feature selection, and data preprocessing to improve model robustness and accuracy. Advanced ensemble approaches, such as RF and gradient boosting decision trees (GBDTs), further enhance detection performance in almonds and walnuts by aggregating multiple trees to improve generalization [38,91]. These models offer a rapid, non-invasive method for mycotoxin screening, improving food safety assessments.

### 4.6. Random Forest

Random forest is a powerful ensemble learning algorithm widely used for mycotoxin detection using hyperspectral imaging due to its high accuracy and robustness against overfitting. It constructs multiple decision trees in parallel and predicts contamination based on majority voting or averaging, effectively handling nonlinear relationships and spectral interactions. RF has been successfully applied to maize and corn classification, distinguishing contaminated from non-contaminated samples with high precision [70,105]. Its effectiveness extends to wheat and barley classification, where it reliably identifies mycotoxin-contaminated grains [47]. By leveraging feature importance metrics, RF enhances interpretability and optimizes mycotoxin detection by identifying key spectral wavelengths. Techniques such as bootstrapping and effective feature selection further improve model accuracy while reducing overfitting.

### 4.7. K-Nearest Neighbors

K-nearest neighbors is a simple yet effective machine learning algorithm frequently used in mycotoxin detection due to its ability to recognize spectral patterns. It operates on instance-based learning, classifying samples based on their proximity to k-nearest neighbors using distance metrics such as Euclidean or Manhattan distance. KNN has been successfully applied to identify mycotoxin contamination in maize, corn, and peanuts using hyperspectral imaging data [70,101,104,106]. However, its performance depends on data quality, optimal k selection, and feature scaling. While computational inefficiency and sensitivity to noise remain challenges, these can be mitigated through cross-validation and preprocessing techniques to improve accuracy and robustness.

### 4.8. Naive Bayes

Naive Bayes is a probabilistic classification algorithm based on Bayes’ theorem, widely used for mycotoxin detection due to its efficiency in handling high-dimensional hyperspectral imaging data. It assumes feature independence, enabling fast classification with minimal data and computational resources. NB has been applied in wheat classification, demonstrating reliable accuracy in predicting mycotoxin contamination from spectral data [57]. It has also been used to classify grains based on fungal infection levels, supporting early mycotoxin detection. While its feature independence assumption may not always hold in HSI data, NB remains effective by leveraging prior probability distributions to enhance classification performance.

### 4.9. Ensemble Boosting

Ensemble boosting is an effective ML technique for quantifying mycotoxins using HSI data. It capitalizes on its ability to improve prediction accuracy by combining multiple weak learners into a strong predictive model. Therefore, it enhances performance in capturing the complex spectral features associated with mycotoxins, which often exhibit subtle variations in HSI data that can be challenging to detect. However, high dimensionality, noise in spectral data, and the need for extensive labeled datasets complicate the trained model [107]. The boosting algorithm addresses these challenges by iteratively focusing on misclassified samples, thus refining its predictions and managing the intricacies of HSI data. Higher prediction accuracy is typically achieved when the model is trained on a diverse dataset that includes a range of mycotoxin concentrations and environmental conditions and when advanced preprocessing techniques are used to reduce noise and enhance spectral quality.

### 4.10. Artificial Neural Network

Artificial neural network models have revolutionized mycotoxin detection in cereals and nuts by leveraging hyperspectral imaging to extract complex spectral–spatial features for precise contamination identification. It effectively models nonlinear relationships within high-dimensional HSI data, enabling the accurate detection of subtle mycotoxin variations. However, ANN requires extensive labeled data and is prone to overfitting, which can be mitigated through techniques like dropout, regularization, and data augmentation.

Deep convolutional neural networks (DCNNs) enhance mycotoxin classification in maize and corn by capturing intricate spectral patterns [103]. Three-dimensional convolutional neural networks (3D-CNNs) further improve aflatoxin detection in peanuts and almonds by preserving spectral dependencies across multiple bands [81,95,104]. Inception and Inception-ResNet architectures optimize feature extraction for contamination identification in almonds and walnuts [86,95], while ResNet, with its deep residual learning framework, enhances classification accuracy for various cereals and nuts. These deep learning models facilitate rapid, non-destructive screening, improving food safety assessments. Their effectiveness depends on comprehensive training datasets that cover diverse mycotoxin types and concentrations, along with robust preprocessing techniques to minimize noise and enhance model reliability.

## 5. Research Scope

Detecting mycotoxins in cereals and nuts remains a major food safety challenge, even with recent progress in machine learning and hyperspectral imaging technologies. HSI has been widely adopted in food processing industries due to its ability to capture chemical information at the molecular level, enabling rapid, non-destructive analysis without the use of harmful chemicals. HSI combined with ML algorithms, such as SVM, RF, ANN, etc., has demonstrated promising results in detecting and quantifying mycotoxins, particularly in cereal grains and nuts.

However, despite these advancements, several challenges persist, including model robustness, accuracy, real-time applicability, feature selection, and regulatory compliance. Improving these aspects is essential for developing scalable, interpretable, and high-performance AI-driven mycotoxin detection systems. Further research is needed to refine feature extraction techniques, enhance classification accuracy, and optimize computational efficiency to ensure practical implementation in industrial food safety workflows.

### 5.1. Enhancing Model Accuracy and Generalization

One of the key challenges in ML-based mycotoxin detection is the generalization of models across different environmental conditions, geographic regions, and sample variations. Current models trained on specific datasets may not perform well when applied to new datasets collected from different storage conditions, humidity levels, or grain varieties [106]. To improve accuracy and generalization, future research should focus on transfer learning and domain adaptation techniques, enabling ML models to adapt to diverse datasets with minimal retraining [60].

Furthermore, combining multi-sensor fusion techniques such as HSI, near-infrared spectroscopy, and fluorescence spectroscopy can provide more robust feature sets for mycotoxin detection. By fusing multiple spectral modalities, ML models can extract more relevant information, reducing false positives and false negatives in mycotoxin classification [56].

### 5.2. Optimized Feature Selection for Mycotoxin Detection

The high-dimensional spectral data generated by HSI and NIR spectroscopy present significant challenges, including computational complexity, redundancy, and the risk of overfitting in ML models. To enhance model efficiency, interpretability, and detection accuracy, effective feature selection techniques are essential [94,96,104]. Future research should focus on hybrid feature selection methods that integrate filter-based, wrapper-based, and embedded approaches [94]. Advanced techniques such as wavelet transforms, PCA, and Non-Negative Matrix Factorization can help extract meaningful spectral features while reducing data dimensionality. To further optimize mycotoxin classification, Genetic algorithms, sparse partial least squares, and principal component regression can be explored for refining spectral feature selection [84,96]. Integrating these strategies will make HSI-based mycotoxin detection models more robust, efficient, and suitable for real-time applications in food safety monitoring.

### 5.3. Integration of Advanced Algorithms

Traditional machine learning methods, such as PLSR, DA, and SVM, rely on handcrafted spectral features, which may not fully capture the intricate spectral–spatial patterns associated with mycotoxin contamination. Deep-learning-based approaches [95], particularly convolutional neural networks (CNNs), recurrent neural networks (RNNs) [38], and vision transformers (ViTs), offer significant advantages by automatically learning hierarchical feature representations from raw hyperspectral data, thereby improving classification accuracy [38]. Hybrid models that integrate deep learning with traditional ML techniques, such as CNN-SVM or 3D-CNN-RF, further enhance detection capabilities by leveraging the strengths of both methodologies. CNNs excel at extracting spatial and spectral features while RNNs effectively capture sequential dependencies in spectral data, enabling the more precise classification and quantification of mycotoxins, especially when their spectral signatures are subtle [108]. Also, recent studies demonstrate that HSI, combined with machine learning, can detect multiple mycotoxins. Kim et al. investigated the feasibility of identifying aflatoxin and fumonisin co-contaminated maize using SWIR imaging and SVM [48]. Similarly, Stasiewicz et al. used a low-cost multispectral sorter to reduce both toxins by 83%, highlighting its practical dual-detection capability [109]. Despite these promising results, the simultaneous detection of multiple mycotoxins remains challenging. Spectral signatures of different toxins often overlap, and their concentrations may be too low relative to other dominant biochemical components to be directly distinguishable. As such, robust chemometric and machine learning models are essential to extract relevant spectral features and improve detection performance.

However, one of the primary barriers to AI adoption in food safety is the lack of model interpretability. Regulatory bodies such as the European Food Safety Authority (EFSA), and the United States Food and Drug Administration (FDA), require transparency in AI-driven mycotoxin detection systems [110]. Future research should prioritize Explainable AI frameworks, including Shapley Additive Explanations (SHAP), Local Interpretable Model-Agnostic Explanations (LIME), and Grad-CAM (Gradient-weighted Class Activation Mapping), to provide clear justifications for model predictions. By advancing hybrid deep learning models and incorporating explainability techniques, AI-driven mycotoxin detection systems can achieve higher accuracy, regulatory compliance, and greater trust among industry stakeholders, ultimately enhancing food safety and agricultural monitoring [57].

### 5.4. High-Dimensional Imbalanced Data

A significant challenge in hyperspectral imaging for mycotoxin detection lies in the limited availability of naturally contaminated samples, which restricts the development and validation of robust detection models. Class imbalance is a fundamental issue, as different contamination classes occur with varying frequencies in real-world scenarios. This uneven distribution can bias model accuracy, favoring more prevalent classes over rare ones [111]. Furthermore, obtaining naturally contaminated samples in a non-destructive manner is often not feasible, necessitating the artificial contamination of cereal grains and nuts. However, replicating all possible mycotoxin concentrations is challenging, creating data deficiencies that can adversely impact machine learning algorithms. Studies like those by [112,113] illustrate this limitation, using only a few concentrations for aflatoxin B1 detection in corn. The characteristics of artificially contaminated data may differ from those of naturally infected samples, complicating model training [114]. To address these issues, future research could explore synthetic data generation and augmentation techniques, such as generative adversarial networks (GANs), enabling models to generalize better for rare contamination cases.

### 5.5. Real-Time Mycotoxin Detection Systems

Most existing ML-based mycotoxin detection methods rely on laboratory-based analysis, limiting their practicality for large-scale industrial applications. To enhance food safety monitoring in grain storage facilities, processing plants, and border inspection checkpoints, there is a growing need for real-time, portable mycotoxin detection systems [72]. Edge computing and real-time processing play a vital role in HSI-based mycotoxin detection, enabling faster decision making in agricultural monitoring and food safety. By processing data closer to the source, these technologies minimize latency and allow for the immediate analysis of hyperspectral data, enhancing efficiency and responsiveness. However, achieving a balance between accuracy and computational efficiency remains a key challenge.

Future research should prioritize the development of lightweight deep learning models optimized for embedded edge devices with limited processing power. Model compression techniques, such as pruning, quantization, and knowledge distillation, can help reduce model size while maintaining detection accuracy. Additionally, integrating IoT sensors with HSI systems can enable real-time data collection, remote monitoring, and automated mycotoxin assessment [70]. Enhancing these capabilities will improve the practicality of AI-driven detection systems, expanding their use in agriculture, food processing, and quality control.

## 6. Conclusions

This review underscores the significant progress in integrating machine learning with hyperspectral imaging for mycotoxin detection in cereals and nuts. HSI has proven to be a powerful, non-destructive imaging tool for detecting contamination at the molecular level, yet challenges persist in managing the vast amounts of redundant spectral data it generates. This highlights the importance of efficient data processing and feature extraction techniques to enhance detection accuracy. Among the ML models reviewed, PLSR has shown an exceptional performance in quantifying mycotoxin levels, particularly for correlated spectra, while SVM effectively mitigates overfitting and DA offers fast, precise classification. In contrast, simpler models like DT and NB struggle with small or collinear datasets. The growing adoption of deep learning has significantly enhanced detection accuracy, particularly with 3D models for hyperspectral images; however, these models often require large training datasets. To address this, pre-trained neural networks are gaining traction, enabling more efficient learning from limited data. Moving forward, optimizing feature selection and refining ML models will be crucial to enhancing the reliability and applicability of mycotoxin detection systems for real-world food safety monitoring.

## Figures and Tables

**Figure 1 toxins-17-00219-f001:**
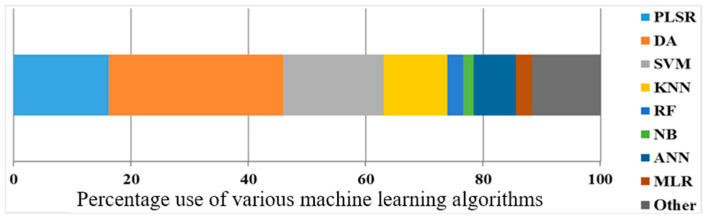
ML algorithms are used to detect mycotoxin using HSI.

**Figure 2 toxins-17-00219-f002:**
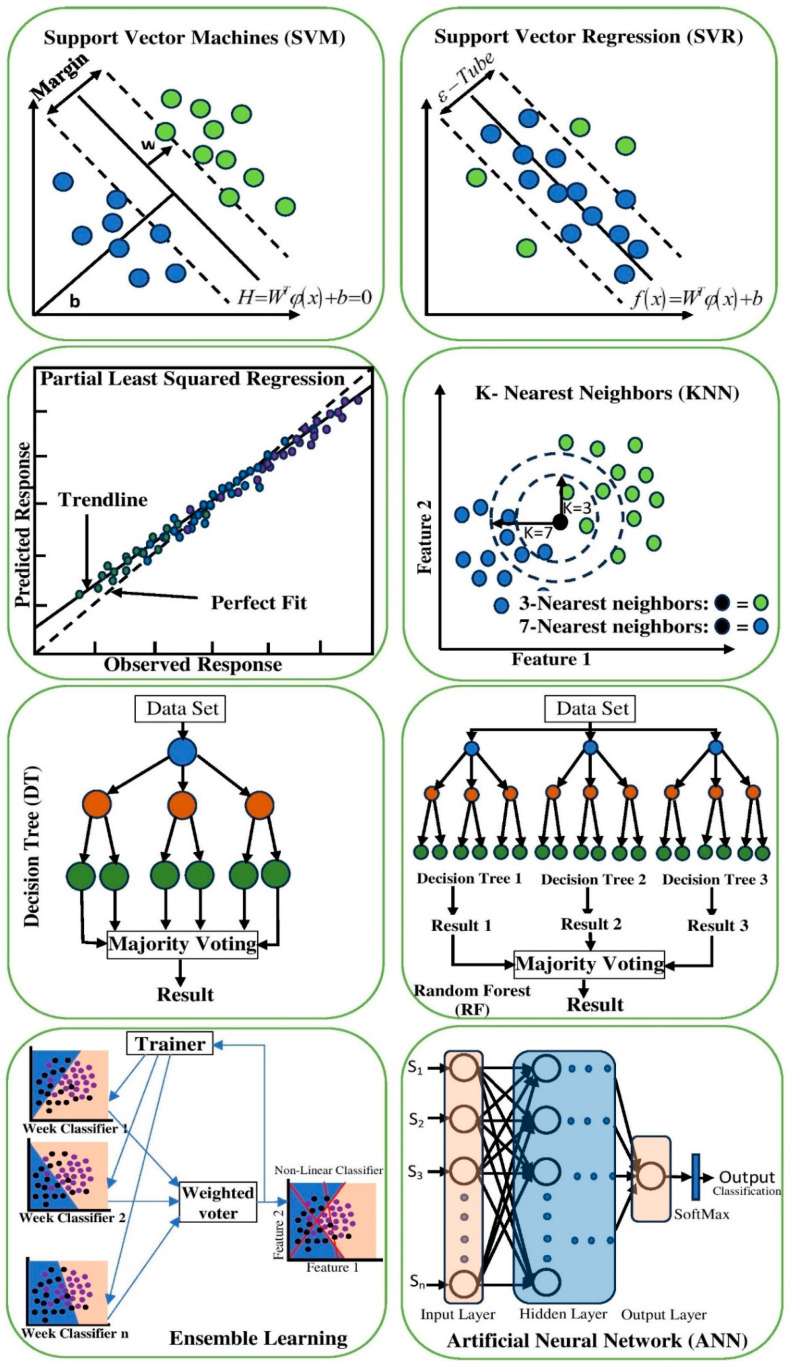
Visual representation of commonly used machine learning algorithms for classification and regression tasks. Top: (1) SVM with margin-based classification, (2) SVR with ε-tube regression. Middle: (3) PLSR for predicted vs. observed fit, (4) KNN by neighbor proximity, (5) DT with majority voting, (6) RF as an ensemble of DTs. Bottom: (7) ensemble learning via weighted weak classifiers, (8) ANN with hidden layer and SoftMax output.

**Table 1 toxins-17-00219-t001:** Most common mycotoxins in cereal grains and nuts and possible health issues.

Mycotoxin	Main Producers	Commonly Affected Foods	Possible Health Issues	MRLs in Cereals and Nuts (µg/kg)
Aflatoxin	*Aspergillus flavus*, *A. parasiticus*	Peanut, pistachio, almond, wheat, corn, and rice	Liver damage, high hemorrhage, DNA damage, impaired body immune function, etc. [8]	Total aflatoxins (nuts): EU: 4–10 [9]; US: 20 [10]
Ochratoxin A	*Aspergillus ochraceus*, *A. carbonarius*	Wheat, corn, and barley	Kidney toxicity in both animals and humans, kidney tumor in rodents [11]	EU: 5–10 (pistachio); 3–5 (cereals) [10,12]
Patulin	*Penicillium* and *Byssochlamys*	Wheat bread	Damaged liver, kidney, immune system, and intestinal tissues [13]	--
T-2 toxin	*Fusarium*, *Myrotecium*	Wheat, barley, rye, oats, and maize	Reduces weight, weakens the immune system [14]	EU: 50 (UP cereal); 150 (UP barley); 100 (UP maize); 1250 (UP oats) [15]
Fumonisins	*Fusarium verticillioides*	Corn and corn-based food products	Can create defects in the neural tube and esophageal cancer [16]	EU: 1000–4000; US: 2000–4000 (cereal) [10]
Ergot alkaloids	*Claviceps* spp.	Wheat, oats, barley, and rye	It can create pulse variation, decrease blood circulation, etc. [17]	EU: 50 (barley, oats, and wheat) [18]
Deoxynivalenol	*Fusarium graminearum*	Wheat, corn, oats, barley, and other cereal grains	It causes vomiting and short-term nausea issues [19]	EU: 150 (P cereal); 1000 (UP cereal), 1500 (UP wheat), 1750 (UP oats) [20]
*Alternaria* toxins	*Alternaria*	Wheat, corn, and other cereal	DNA strands can be broken, reducing cell proliferation [21]	EU (recommended in cereal, not MRLs): 2 (Alternariol); 500 (Tenuazonic acid) [22]
Zearalenone	*Fusarium graminearum* and *F. culmorum*	Wheat, rice, oats, barley, and rye	Can have an impact on hormonal balance [5]	EU: 100–350 (cereal) [10]

MRLs (maximum residue limits); EU (European Union); US (United States); P (processed); UP (unprocessed).

**Table 2 toxins-17-00219-t002:** ML techniques to detect mycotoxins in wheat using hyperspectral images.

Mycotoxin	Sample Number	Wavelength (nm)	Preprocessing	ML Methods	Performance	Reference
DON	600	1000–16,000	SNV, MSC	LDA	Acc: 83%	[56]
DON	72	250–2500	SG, MSC	RF, SVM, CNN	Acc: 98.95%	[57]
DON	300	900–1700	MSC, SG, 1st and 2nd dev, SNV	PLSR	R^2^: 0.88	[54]
DON	120	970–1623	MSC, 1st and 2nd dev, SNV	PLSR, SVR	R^2^: 0.88	[49]
DON	480	960–1700	Raw	KNN	Acc: 80%	[39]
DON	50	900–1700	MSC, 1st and 2nd dev, SNV	LDA, NB, KNN	Acc: 100%	[55]
DON	96	400–2500	SNV, MSC	SVM, SAE	Acc: 96%	[58]
DON	150	900–1700	MSC, 1st and 2nd dev, SNV	PLSR, LDA	Acc: 62.7%	[59]
AFB1	150	899–1725	SG, SNV	PLSR	R^2^: 0.99	[60]
OTA	20 kg	1000–1600	Raw	DA	Acc: 99.8%	[61]
OTA	10 kg	1000–1600	Raw	DA	Acc: 100%	[46]

DON (deoxynivalenol); A**F**B1 (aflatoxin B1); OTA (ochratoxins); SNV (standard normal variate); MSC (multiplicative scatter correction); SG (Savitzky–Golay); LDA (linear discriminant analysis); SVM (support vector machine); PLSR (partial least squares regression); SVR (support vector regression); KNN (k-nearest neighbors); RF (random forest); CNN (convolutional neural network); NB (Naive Bayes); SAE (sparse autoencoder).

**Table 3 toxins-17-00219-t003:** ML techniques to detect mycotoxin for corn using hyperspectral images.

Mycotoxin	Sample Number	Wavelength (nm)	Preprocessing	ML Methods	Performance	Reference
AFB1	900	786–824	SG	LDA	Acc: 89.6%	[62]
AFB1	958	380–814	SNV	SVR	Acc: 98%	[63]
AFB1	228	900–2500	SNV, SG	SVM, PLS-DA	Acc: 95.7%	[48]
AFB1	968	327–1098 930–2548	Raw	PLSR	R^2^: 0.924	[64]
AFB1	26	400–1000	SNV	SVM, PLSR	R^2^: 0.95	[65]
AFB1	300	900–2500	SNV, SG, 1st and 2nd dev	PLS-DA	Acc: 90.4%	[40]
AFB1	320	1000–2500	SNV	PLSR, SVR	R^2^: 0.97	[66]
AFB1	122	785	SG, SNV	SVM	Acc: 95.7%	[67]
AFB1	958	460–929	SNV, MSC	KNN, SVM, LDA	Acc: 87.3%	[68]
AFB1	250	1100–2000	SG, 1st and 2nd dev	LDA	Acc: 95.56%	[69]
AFB1	6	292–865	MSC, PCA,	RF, KNN	Acc: 99.38%	[70]
DON	98	893–1730	MSC, SNV, SG	RF, NN, LR, KNN	Acc: 98.6% R^2^: 81.9	[71]
DON	176	900–1700	SNV, SG smoothing	PLSR, PLS-DA	R^2^: 0.974, Acc: 98.8%	[72]
FM	58	1000–2100	SNV	PLSR	R^2^: 0.98	[73]
ZEN	1890	400–1000	Raw	SVM, DA, PLSR, NN	Acc: 93.33 %	[74]

DON (deoxynivalenol); A**F**B1 (aflatoxin B1); FM (fumonisins); ZEN (zearalenone); SNV (standard normal variate); MSC (multiplicative scatter correction); SG (Savitzky–Golay); PCA (principal component analysis); LDA (linear discriminant analysis); SVM (support vector machine); PLSR (partial least squares regression); SVR (support vector regression); RF (random forest).

**Table 4 toxins-17-00219-t004:** ML techniques to detect mycotoxin for various cereal grains using hyperspectral images.

Other Cereal	Mycotoxin	Sample Number	Wavelength (nm)	Preprocessing	ML Methods	Performance	Reference
Barley	DON	590	367–1048	MAF	CNN	Acc: 89.81%	[75]
Barley	DON	1004	382–1030	Raw	PLDA	R^2^: 0.931	[76]
Barley	OTA	101 kg	1000–1600	Raw	DA	Acc: 100%	[77]
Oats	DON	119	900–1700	SNV, 1st dev	PLSR, RF, KNN, NB	Acc: 77.8%	[47]
Oats	DON	220	1000–2500	Normalization	PLSR, LDA	R^2^: 0.8	[78]
Oats	T-2	119	900–1700	SNV, 1st dev	PLSR, RF, KNN, NB	Acc: 94.1%	[47]

DON (deoxynivalenol); OTA (ochratoxins); T-2 (T-2 toxin); SNV (standard normal variate); MAF (moving-average filtering); LDA (linear discriminant analysis); PLDA (partial least squares discriminant analysis); PLSR (partial least squares regression); KNN (k-nearest neighbors); RF (random forest); CNN (convolutional neural network); NB (Naive Bayes); DA (discriminant analysis).

## Data Availability

No new data were created or analyzed in this study.

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
