# Peer review of "Detection of Mycotoxins in Cereal Grains and Nuts Using Machine Learning Integrated Hyperspectral Imaging: A Review"

_toxins, 2025, doi:10.3390/toxins17050219_

Round 1

Reviewer 1 Report

Comments and Suggestions for Authors

This review have explored hyperspectral imaging (HSI) integrated with machine learning (ML) algorithms as a promising approach for detecting and quantifying mycotoxins in cereal grains and nuts.  The paper was well written and some minor problem should be revised. 

1, Table 1 have listed most common mycotoxins in cereal grains and nuts and possible health issues. The MRLs in some typical foods should be added.
2, In Table 2-5, ML techniques to detect mycotoxin for wheat, corn nuts and various other cereal grains using hyperspectral images have been conluded. More infomarion, such as sample numbers, toxic concentration ranges, also should be supplemented and compared. 
3. Can HSI detect several mycotoxins simoutanlously ? More discussion should be added.

Author Response

This review have explored hyperspectral imaging (HSI) integrated with machine learning (ML) algorithms as a promising approach for detecting and quantifying mycotoxins in cereal grains and nuts.  The paper was well written and some minor problem should be revised. 

Q1) Table 1 have listed most common mycotoxins in cereal grains and nuts and possible health issues. The MRLs in some typical foods should be added.

Answer: Thanks for your valuable suggestion. We added a column in table of ‘MRLs in food”.

Q2) In Table 2-5, ML techniques to detect mycotoxin for wheat, corn nuts and various other cereal grains using hyperspectral images have been conluded. More infomarion, such as sample numbers, toxic concentration ranges, also should be supplemented and compared. 

Answer: Thank you for your valuable suggestion. In response, we added the number of samples used in each individual study to the revised manuscript.

Q3) Can HSI detect several mycotoxins simoutanlously ? More discussion should be added.

Answer: Thank you for your insightful comment. We have revised the manuscript to include a more detailed discussion on the potential and limitations of hyperspectral imaging for the simultaneous detection of multiple mycotoxins. We added the following discussion in section 3.3 I of the revised manuscript.

“Also, recent studies demonstrate that HSI, combined with machine learning, can detect multiple mycotoxins. Kim et al. investigated the feasibility of identifying aflatoxin and fumonisin co-contaminated maize using SWIR imaging and SVM [42]. Similarly, Stasiewicz et al. used a low-cost multispectral sorter to reduce both toxins by 83%, highlighting its practical dual-detection capability[103]. Despite these promising results, the simultaneous detection of multiple mycotoxins remains challenging. Spectral signatures of different toxins often overlap, and their concentrations may be too low relative to other dominant biochemical components to be directly distinguishable. As such, robust chemometric and machine learning models are essential to extract relevant spectral features and improve detection performance.”

Reviewer 2 Report

Comments and Suggestions for Authors Some sentences are overly complex and could be rewritten for better readability. Simplifying sentence structure, particularly in the abstract and introduction, would enhance accessibility for readers unfamiliar with hyperspectral imaging.

Figures illustrating ML model architectures for hyperspectral image analysis would add visual clarity to the manuscript.

The authors should explicitly outline the contributions in the introduction.

Adding discussions on novel techniques such as hybrid deep learning models would enhance the completeness of the review.

Author Response

Some sentences are overly complex and could be rewritten for better readability. Simplifying sentence structure, particularly in the abstract and introduction, would enhance accessibility for readers unfamiliar with hyperspectral imaging.

Answer: Thank you for your helpful feedback. We carefully reviewed and updated the revised manuscript.

Figures illustrating ML model architectures for hyperspectral image analysis would add visual clarity to the manuscript.

Answer: Thank you for your suggestion. A figure illustrating various machine learning model architectures for hyperspectral image analysis was already included in the original manuscript (Figure 2). To improve clarity, we have revised the figure caption and cross-referenced it more explicitly within the relevant discussion in the text. We believe this will help readers better understand the model structures and their application in the study.

The authors should explicitly outline the contributions in the introduction.

Answer: Thank you for the helpful feedback. In response, we have revised the introduction to outline the key contributions of the research. In the revised manuscript, a dedicated paragraph has been added toward the end of the introduction, summarizing the key contributions of the study.

Adding discussions on novel techniques such as hybrid deep learning models would enhance the completeness of the review.

Answer: Thank you for the insightful suggestion. In response, we have updated the manuscript to include a dedicated discussion on emerging hybrid deep learning techniques. This section highlights how combining convolutional neural networks with traditional machine learning algorithms or other deep learning modules (e.g., transformer, attention mechanisms) can improve the performance and generalizability of hyperspectral image analysis. We have also cited recent studies demonstrating the effectiveness of these hybrid models in food safety applications, including mycotoxin detection.

Reviewer 3 Report

Comments and Suggestions for Authors

This is a relevant manuscript regarding the comprehensive review of analytical methodologies for the determination of mycotoxins in cereal grains and nuts. Nonetheless, minor suggestions/revisions should be addressed as described in the next comments. 

Comment 1: Lines 126 to 128 – “In the review planning the objectives, scope, and criteria were settled to ensure the structure and analysis process.”. The authors should clearly present these objectives, scope and the criteria established for article selection

Comment 2: What do the authors mean by this: “We used various combinations of the keywords ‘mycotoxin’, ‘hyperspectral imaging’, and ‘machine learning’ to find the relevant research paper.”. The strings used for database source should be given (at least in supplementary material).

Comment 3: Since the paper is focused on specific matrices, namely nuts and cereal grains, why was not this considered on the search string? Or was this considered as an “exclusion criteria” after screening of the 83 papers obtained?

Comment 3: Lines 149 – 150 “In this way, a few of the screened papers are considered for further description in the later section.”. How many papers (number) were screened and used for data analysis; how many were excluded and why?

Comment 4: Section 2. Research Methodology needs improvement. It is focused on theory regarding search methods, and the actual procedure is lost in there. The authors should keep it simpler, and straightforward to what was done. Meaning: (1) define objectives, scope, and criteria of the search; (2) keywords and strings used, as well as the databases (including dates of search); (3) How many articles were obtained from the search and how many were included for final analysis.

Comment 5: The title should be changed to “Search methodology” or “Search strategy” not “research methodology”

Comment 6: “This mycotoxin creates a huge threat to human health.” (line 155) – Something is missing in this sentence.

Comment 7: Section 3 name does not relate to the actual content of this section. I would advise to change it to “3. Hyperspectral detection of mycotoxins in cereal and nuts” and move the first part (lines 153 to 157) to Introduction section, including Table 1. And subsection 3.1. would be “Detection of Mycotoxins in Wheat” and so on.

Comment 8: Lines 181 to 182 refer to an HSI article for pears. The focus of the article is nuts and cereal grains. Can the authors change this example for one within the scope of the article?

Comment 9: The data given in Figure 1. is based on the articles retrieve from the initial search? If so, the authors should provide the number of articles considered to come to this conclusion: “The percentage usage of the most common ML algorithms for mycotoxin detection using HSI is illustrated in Figure 1.”. Because this conclusion is dependent on the authors’ article selection criteria and the articles chosen.

Comment 10: In line 336, the authors refer to “percentage”, and on the Figure caption “Percentile”. Please provide the correct measurement.

Comment 11: Please provide in each table a footnote with the description of the abbreviations used on the corresponding table.

Comment 12: Line 520: “Mycrotoxin Detection” to “Mycotoxin detection”

Comment 13: “A significant challenge in HSI for mycotoxin detection is the scarcity of naturally contaminated samples and the difficulty of achieving uniform contamination levels through artificial means.”. Can the authors explain this statement? It should be revised, since in, one hand, naturally contaminated samples are quite high in terms of mycotoxin prevalence for cereals and nuts and, on the other hand, “uniform contamination levels” can be achieved by spiking the target matrices at the same concentration levels in replicate (the difficulty here might be in finding blank samples).

Author Response

This is a relevant manuscript regarding the comprehensive review of analytical methodologies for the determination of mycotoxins in cereal grains and nuts. Nonetheless, minor suggestions/revisions should be addressed as described in the next comments. 

Comment 1: Lines 126 to 128 – “In the review planning the objectives, scope, and criteria were settled to ensure the structure and analysis process.”. The authors should clearly present these objectives, scope and the criteria established for article selection

Answer: Thanks for your valuable feedback. We reviewed and rewrote section 2 in the revised manuscript.

Comment 2: What do the authors mean by this: “We used various combinations of the keywords ‘mycotoxin’, ‘hyperspectral imaging’, and ‘machine learning’ to find the relevant research paper.”. The strings used for database source should be given (at least in supplementary material).

Answer: Thanks for your valuable feedback. We reviewed and rewrote section 2 in the revised manuscript.

Comment 3: Since the paper is focused on specific matrices, namely nuts and cereal grains, why was not this considered on the search string? Or was this considered as an “exclusion criteria” after screening of the 83 papers obtained?

Answer: Thank you for this important observation. In our review methodology, the focus on nuts and cereal grains was applied as an inclusion/exclusion criterion during the screening stage, rather than being explicitly embedded in the initial search string. The broader search string was intentionally designed to capture a wide range of studies related to mycotoxin detection using hyperspectral imaging and machine learning.

Comment 3: Lines 149 – 150 “In this way, a few of the screened papers are considered for further description in the later section.”. How many papers (number) were screened and used for data analysis; how many were excluded and why?

Answer: Thanks for your valuable feedback. We reviewed and rewrote section 2 in the revised manuscript.

Comment 4: Section 2. Research Methodology needs improvement. It is focused on theory regarding search methods, and the actual procedure is lost in there. The authors should keep it simpler, and straightforward to what was done. Meaning: (1) define objectives, scope, and criteria of the search; (2) keywords and strings used, as well as the databases (including dates of search); (3) How many articles were obtained from the search and how many were included for final analysis.

Answer: Thanks for your valuable feedback. We reviewed and rewrote section 2 in the revised manuscript.

Comment 5: The title should be changed to “Search methodology” or “Search strategy” not “research methodology”

Answer: Thanks for your suggestion. We revised and changed the title ‘research methodology’ into “Search methodology”.

Comment 6: “This mycotoxin creates a huge threat to human health.” (line 155) – Something is missing in this sentence.

Answer: Thanks for your feedback. We revised and updated a couple of sentences as follows: ‘Some mycotoxins are highly toxic and carcinogenic, and certain types cannot be eliminated from food through heating, cooking, or other conventional processing methods. Continuous consumption of these mycotoxins poses a significant health risk to both humans and animals.’

Comment 7: Section 3 name does not relate to the actual content of this section. I would advise to change it to “3. Hyperspectral detection of mycotoxins in cereal and nuts” and move the first part (lines 153 to 157) to Introduction section, including Table 1. And subsection 3.1. would be “Detection of Mycotoxins in Wheat” and so on.

Answer: Thanks for your suggestion. We revised the manuscript and changed the title of section 3 to ‘Hyperspectral Analysis of Mycotoxins in Cereal and Nuts’. Also, moved the first part in the introduction section, including table 1.

Comment 8: Lines 181 to 182 refer to an HSI article for pears. The focus of the article is nuts and cereal grains. Can the authors change this example for one within the scope of the article?

Answer: Thank you for your valuable suggestion. Since we were unable to find a suitable alternative reference within the scope of nuts and cereal grains, we have removed the example related to pears in the revised manuscript.

Comment 9: The data given in Figure 1. is based on the articles retrieve from the initial search? If so, the authors should provide the number of articles considered to come to this conclusion: “The percentage usage of the most common ML algorithms for mycotoxin detection using HSI is illustrated in Figure 1.”. Because this conclusion is dependent on the authors’ article selection criteria and the articles chosen.

Answer: Thank you for your insightful comment. Yes, the data presented in Figure 1 is based on 83 research articles that utilized hyperspectral imaging and machine learning for mycotoxin detection in cereal grains and nuts, published between 2015 and 2024. We have clarified this point and included the number of articles and selection criteria in the revised manuscript.

Comment 10: In line 336, the authors refer to “percentage”, and on the Figure caption “Percentile”. Please provide the correct measurement.

Answer: Thanks for your valuable feedback. We revised and corrected the figure caption and used ‘Percentage’

Comment 11: Please provide in each table a footnote with the description of the abbreviations used on the corresponding table.

Answer: Thank you for your valuable suggestion. We have revised the manuscript and added footnotes to each table, providing clear descriptions of all abbreviations used.

Comment 12: Line 520: “Mycrotoxin Detection” to “Mycotoxin detection”

Answer: Thank you for your valuable feedback. The word has been revised, and the spelling has been corrected in the revised manuscript.

Comment 13: “A significant challenge in HSI for mycotoxin detection is the scarcity of naturally contaminated samples and the difficulty of achieving uniform contamination levels through artificial means.”. Can the authors explain this statement? It should be revised, since in, one hand, naturally contaminated samples are quite high in terms of mycotoxin prevalence for cereals and nuts and, on the other hand, “uniform contamination levels” can be achieved by spiking the target matrices at the same concentration levels in replicate (the difficulty here might be in finding blank samples).

Answer: Thanks for your valuable feedback. We revised and updated the manuscript.